# The Introduction of Detergents in Thermal Proteome Profiling Requires Lowering the Applied Temperatures for Efficient Target Protein Identification

**DOI:** 10.3390/molecules28124859

**Published:** 2023-06-20

**Authors:** Yuying Ye, Kejia Li, Yanni Ma, Xiaolei Zhang, Yanan Li, Ting Yu, Yan Wang, Mingliang Ye

**Affiliations:** 1CAS Key Laboratory of Separation Science for Analytical Chemistry, National Chromatographic R&A Center, Dalian Institute of Chemical Physics, Chinese Academy of Sciences (CAS), Dalian 116023, China; yuyingye1809@dicp.ac.cn (Y.Y.); lkj2016@dicp.ac.cn (K.L.); ynma@dicp.ac.cn (Y.M.); xiaolei16@dicp.ac.cn (X.Z.); liyanan@dicp.ac.cn (Y.L.); yuting@dicp.ac.cn (T.Y.); wy805@dicp.ac.cn (Y.W.); 2University of Chinese Academy of Sciences, Beijing 100049, China

**Keywords:** mild detergents, thermal proteome profiling, target protein identification

## Abstract

Although the use of detergents in thermal proteome profiling (TPP) has become a common practice to identify membrane protein targets in complex biological samples, surprisingly, there is no proteome-wide investigation into the impacts of detergent introduction on the target identification performance of TPP. In this study, we assessed the target identification performance of TPP in the presence of a commonly used non-ionic detergent or a zwitterionic detergent using a pan-kinase inhibitor staurosporine, our results showed that the addition of either of these detergents significantly impaired the identification performance of TPP at the optimal temperature for soluble target protein identification. Further investigation showed that detergents destabilized the proteome and increased protein precipitation. By lowering the applied temperature point, the target identification performance of TPP with detergents is significantly improved and is comparable to that in the absence of detergents. Our findings provide valuable insight into how to select the appropriate temperature range when detergents are used in TPP. In addition, our results also suggest that the combination of detergent and heat may serve as a novel precipitation-inducing force that can be applied for target protein identification.

## 1. Introduction

Drug target protein identification is essential to understand the mechanisms of action of drugs [1], to explain drug polypharmacology [2], and to avoid undesired side effects [3,4]. Proteome-wide target identification methods can be primarily classified into two categories: modification-based methods and modification-free methods [5]. Modification-based methods, including affinity purification [4] and photoaffinity labeling [6], require prior drug modification to isolate target proteins from the background proteome for target identification. Different from modification-based methods, modification-free methods bypass the laborious drug modification and can directly identify target proteins by the proteome-level characterization of protein biophysical properties [7]. Proteins exhibiting altered biophysical properties upon drug treatment are regarded as putative drug target proteins [8,9,10,11]. Among the modification-free methods, thermal proteome profiling (TPP) [8] is the most widely applied method for drug target identification. TPP is based on the concept that ligand binding can increase protein thermal stability leading them to be more resistant to heat-induced protein precipitation. As a result, the target proteins will display an altered abundance in the soluble fractions.

TPP has demonstrated excellent performance in the identification of drug target proteins from the soluble proteome. For instance, several labs [8,12,13] including our lab [14,15,16] have effectively applied TPP to identify dozens of kinase target proteins for a pan-kinase inhibitor, staurosporine. However, as TPP relies on protein precipitation for target protein identification, it cannot deal with proteins that exist as cellular precipitates such as membrane proteins, which account for 80% of drug targets [17]. To make TPP applicable to membrane proteins, researchers introduced non-ionic detergents in the cell lysis buffer to solubilize membrane proteins for membrane target protein identification [18]. Although the combination of non-ionic detergents and TPP has been a common practice to improve the proteome [18,19,20], there is no systematic investigation of the impact of introducing detergents on TPP target identification performance.

In this study, we investigated the impact of two mild detergents (including both non-ionic and zwitterionic) on the target identification performance of TPP. Specifically, we used a pan-kinase inhibitor staurosporine and compared the identified kinase targets in the presence of Nonidet P-40 (NP40) or 3-cholamidopropyl dimethylammonio 1-propanesulfonate (CHAPS) with those in the absence of any detergent. We first used a single-temperature version of TPP for this comparison. By heating the cell lysates to 52 °C, the temperature that we have previously validated in our study as optimal for target protein identification without detergents [14], we observed that the addition of either of these detergents substantially reduced the number of identified kinase targets. We further demonstrated that the impaired performance was due to both the detergents having decreased protein thermal stability. Directed by the results, we used a lower temperature and evaluated the target identification performance in the presence of a non-ionic detergent or a zwitterionic detergent. Reassuringly, the number of identified kinase targets was remarkably improved, and was comparable to that identified without detergents at 52 °C. A very recent study [21] also reported that the non-ionic detergent NP40 can modify protein thermal stability, and they therefore concluded that the introduction of detergents may not be compatible with the principles of TPP for target identification. Different from that conclusion, our results underscore the importance of using a lower temperature range in TPP for efficient target identification when utilizing detergents (either non-ionic or zwitterionic).

## 2. Results

### 2.1. Comparing the Proteomes Extracted Using Different Detergents

It is well recognized that detergents can increase the solubility of membrane proteins. However, it is unknown whether the different types of detergents have different solubilization preferences. We first compared the proteome extracted without detergents to those extracted with 0.4% NP40 (*v*/*v*) or 1% CHAPS (*wt*/*v*). NP40 is a non-ionic detergent with an uncharged head group, while CHAPS is a zwitterionic detergent that contains both negatively and positively charged atomic groups, resulting in an overall neutral charge (Figure 1A). The applied dosage of the non-ionic detergent NP40 was determined according to the previously reported value that has been used in TPP for membrane protein target identification [18]. For the zwitterionic detergent CHAPS, we used a concentration of 1%, which has been shown to be unable to solubilize heat-induced protein precipitation [18].

Not surprisingly, the addition of either of the detergents increased the number of extracted proteins. The mean values of three replicates were as follows: PBS (4889, standard deviation 78), NP40 (5371, standard deviation 96), and CHAPS (5401, standard deviation 80) (Figure 1B). Moreover, the magnitudes of the increases were consistent across the two applied detergents (~10%). A principal component analysis (PCA) of protein abundances in the three proteomes shows that proteomes extracted using two detergents exhibit similarity while being distinctly separate from the proteome extracted using PBS (Figure 1C). We next evaluated the overlaps of the three identified proteomes (Figure 1D). The largest portion is the proteins that were identified under all conditions. Notably, a substantial portion of proteins exclusive to the detergent-extracted proteomes exhibit significant overlap, suggesting that non-ionic and zwitterionic detergents possess similar solubilization properties towards the proteome. Gene Ontology (GO) analysis of the 638 proteins that were not detected in the PBS-extracted proteome but were present in both NP40- and CHAPS-extracted proteomes showed strong enrichment in membrane-related GO terms (Figure 1E).

In summary, these results showed that the mild detergents, either non-ionic or zwitterionic, can moderately (~10% in our data) increase the extracted proteome by primarily increasing the solubilization of membrane proteins. In addition, no obvious difference in solubilization preference was observed between the non-ionic detergent NP40 and zwitterionic detergent CHAPS.

### 2.2. The Addition of Mild Detergents Remarkably Impaired the Target Identification Performance of TPP-52 °C

Having demonstrated that the addition of mild detergents can increase the extracted proteome, we next evaluated whether it influences the target identification performance of TPP. Staurosporine is a pan-kinase inhibitor and has been widely used to evaluate the performance of existing target-identifying methods [8,12,13,14]. Moreover, although staurosporine is known to bind kinase family proteins, the kinase proteome has the same wide range of thermal stabilities as the whole proteome, albeit the kinase proteome has a more concentrated distribution (Appendix A), indicating that the kinase proteome could represent the whole proteome to some extent. Therefore, we also applied staurosporine for this evaluation (Figure 2A). Specifically, K562 cell lysates (extracted using PBS or PBS/0.4% NP40 or PBS/1% CHAPS) were incubated with 20 μM staurosporine or an equal volume of vehicle (Dimethylsulfoxide, abbreviated as DMSO) at room temperature for 30 min, followed by heating at 52 °C for 3 min. All experiments were performed in at least three technical replicates. The drug-bound target proteins are more resistant to heat-induced precipitation than the unbound targets and thus will display a higher abundance in the supernatants. To determine drug-binding proteins, the proteins remaining in the supernatants were collected and then subjected to bottom-up LC-MS/MS analysis in data-independent acquisition (DIA) mode. Protein quantification was achieved by Spectronaut using the directDIA module.

The peptide quantities across replicates between drug-treated and control groups in the PBS-TPP 52 °C, NP40-TPP 52 °C, and CHAPS-TPP 52 °C experiments all display correlations of >0.99 (Figure 2B), indicating the excellent reproducibility of our technical replicates and DIA quantification. Additionally, the median value of the coefficient of variation for peptide quantities among replicates in each experiment is consistently <0.1 (Figure 2C), further confirming the high reproducibility. We then compared the kinase targets identified by PBS-TPP at 52 °C, NP40-TPP at 52 °C, and CHAPS-TPP at 52 °C. Considering that drug binding typically increases the stability of target proteins [8], we only considered proteins that displayed increased stabilities as candidate target proteins. By using a significance cutoff of −log_10_
*p*-value > 2 and log_2_FC > 0, we identified 47 kinase targets using PBS-TPP at 52 °C (Figure 2D). In contrast, the introduction of 0.4% NP40 resulted in the identification of only 21 kinase targets, and only 29 kinase targets were identified when we introduced 1% CHAPS, although it is noteworthy that most of the kinase targets identified by PBS-TPP at 52 °C were also present in the NP40- and CHAPS-TPP 52 °C datasets (Figure 3A,B). Additionally, although the NP40-TPP 52 °C experiment involved three technical replicates, while the PBS-TPP 52 °C and CHAPS-TPP 52 °C experiments had four replicates, we found that there was almost no difference in the performance of kinase target identification between performing three and four technical replicates (Appendix A). Therefore, we can conclude that the difference observed in the performance of detergent TPP and detergent-free TPP cannot be attributed to the variation in the number of technical replicates.

We next analyzed the reasons for the failure of identifying some kinase targets in the presence of detergents. We first normalized the protein intensities in PBS-TPP 52 °C dataset and detergent-TPP 52 °C datasets using the quantile method [22] (R limma package). In the NP40-TPP 52 °C dataset, 31 kinase proteins were identified as targets by PBS-TPP at 52 °C but were not identified as targets by NP40-TPP at 52 °C (Figure 3A). In both the control and staurosporine-treated groups, we observed that the majority of kinase proteins exhibited lower intensities in the NP40-TPP 52 °C dataset compared to the PBS-TPP 52 °C dataset (Figure 3C). Importantly, in the staurosporine-treated group, this reduction was particularly pronounced. Moreover, we also found the same trend for CHAPS-TPP 52 °C (Figure 3D). This result indicates that the presence of detergents increases the tendency of these kinase proteins to precipitate, thereby counteracting the precipitation difference between control and drug-treated groups.

### 2.3. Both Non-Ionic and Zwitterionic Detergents Decrease Protein Thermal Stabilities

Inspired by the above results, we next investigated whether the applied detergents could impact the proteins’ thermal stabilities. To this end, K562 cell lysates, prepared using detergent-free or detergent (PBS/0.4% NP40 or PBS/1% CHAPS) lysis buffers, were subjected to heating with a thermal gradient ranging from 40 °C to 80 °C, and the resulting supernatant fractions were collected through centrifugation.

Analysis of the supernatant proteome using SDS-PAGE (Sodium Dodecyl Sulfate Polyacrylamide Gel Electrophoresis) revealed that the proteome extracted using detergents exhibited lower thermal stabilities compared to the proteome extracted using only PBS, although the degree of reduction varied across different detergents (Figure 4A,B). Drewes and co-workers [18] used tandem mass tag (TMT)-labeling-based quantitative proteomics showing that the presence of 0.4% NP40 decreased protein thermal stability by 2.9 °C. In close agreement with the reported MS-level result, our two independent SDS-PAGE experiments yielded similar results, demonstrating a decrease in protein thermal stability by 2.6 °C and 2.2 °C, respectively, upon the introduction of 0.4% NP40 (Figure 4B and Appendix A). Moreover, our results from these SDS-PAGE experiments consistently revealed that the zwitterionic detergent CHAPS has a greater destabilizing effect on protein thermal stability compared to the non-ionic detergent, i.e., 0.4% NP40 (Figure 4B and Appendix A).

### 2.4. Using a Lower Temperature Can Restore the Target Identification Performance

Guided by the above results, we wondered if using a lower temperature could improve the target identification performance in the presence of detergents. To test this, we first applied TPP at 49 °C and compared its target identification performance in the presence and absence of 0.4% NP40 (non-ionic detergent). By using a significance cutoff of −log_10_
*p*-value > 2 and log_2_FC > 0, we identified 69 candidate target proteins in the NP40-TPP 49 °C experiment, among which 42 were kinase target proteins (Figure 5A). This kinase target number is close to that identified by PBS-TPP at 52 °C (47 kinase target proteins). By contrast, consistent with our previous report [14], lowering the heating temperature to 49 °C significantly impaired the target-identification performance of PBS-TPP: 25 kinase targets out of 52 candidate target proteins were identified by PBS-TPP at 49 °C (Figure 5B), compared to 47 kinase targets out of 59 candidate target proteins identified by PBS-TPP at 52 °C (Figure 2D).

Having validated the improved performance of TPP with non-ionic detergent by lowering the applied temperature, we next explored whether this principle could also be applicable to the zwitterionic detergent CHAPS, which has been shown to remarkably impact protein thermal stability (Figure 4B). We first also applied 49 °C for CHAPS-TPP. Reassuringly, we identified 41 kinase target proteins for CHAPS-TPP at 49 °C (Figure 5C), which is comparable to 47 kinase target proteins for PBS-TPP at 52 °C. Notably, we observed that 44% of the kinase targets identified by CHAPS-TPP at 49 °C were not identified as staurosporine targets by PBS-TPP at 52 °C (Figure 5D), indicating these two conditions are highly complementary in identifying target proteins. When we further lowered the applied temperature to 47 °C, we found that the number of identified kinase targets (41 kinase target proteins) remained consistent with that at 49 °C (Figure 5E). However, the kinase targets identified by CHAPS-TPP at 47 °C display a larger overlap with those identified by PBS-TPP at 52 °C (Figure 5F) than CHAPS-TPP at 49 °C, indicating that CHAPS-TPP at 47 °C more closely resembles the precipitation conditions in PBS-TPP at 52 °C. This observation is consistent with our SDS-PAGE result, which showed that CHAPS destabilized the proteome by about 7 °C (Figure 4B and Appendix A).

Unfortunately, among the 18 additional kinase targets identified by CHAPS-TPP at 49 °C and the 13 additional kinase targets identified by CHAPS-TPP at 47 °C, none were found to be membrane proteins (Appendix A). This can be explained by the fact that only about 13% of kinase proteins are membrane proteins (Appendix A) and membrane proteins are often in lower abundance compared to cytosolic proteins [23].

Except from the membrane proteins, we want to know if certain types of target proteins are preferably identified by the zwitterionic detergent CHAPS TPP instead of detergent-free TPP. To this end, we selected the kinase target proteins that were stably identified as staurosporine targets in all CHAPS-TPP datasets (at 52 °C, 49 °C, and 47 °C) but were not identified as staurosporine targets in either the PBS-TPP 52 °C dataset or the PBS-TPP 49 °C dataset. This leads to eight kinase proteins (Figure 6A). All eight of these kinase proteins were also present in the PBS-TPP datasets (Appendix A) but were not identified as staurosporine targets by PBS-TPP. Notably, three 5’-AMP-activated protein kinase subunits were found among these eight proteins. A protein–protein interaction analysis with Bioplex [24] revealed that these three proteins interact with each other, existing as a protein complex in human cells. It is possible that the presence of zwitterionic detergent destabilized the protein complex, increasing the accessibility of the subunits to staurosporine binding. By contrast, using the same strategy to screen kinase proteins that were stably identified as staurosporine targets by PBS-TPP but not identified in any of the CHAPS-TPP datasets, we obtained six kinase target proteins (Figure 6B). No reported interactions were found among these target kinase proteins. In summary, our results demonstrate that beyond membrane targets, the presence of detergent may contribute to identifying target proteins from protein complexes that would otherwise be missed by detergent-free TPP.

## 3. Discussion

The introduction of non-ionic detergent NP40 to TPP has been demonstrated to be able to alter protein thermal stability in previous studies [18,21]. However, there is no definitive conclusion on whether the introduction of detergents impacts the target identification performance of TPP. To answer this question, we systemically investigated the influences of both the non-ionic and zwitterionic detergents on the number of identified kinase target proteins for a pan-kinase inhibitor. Our data revealed that the introduction of either non-ionic or zwitterionic detergent significantly decreased the number of kinase targets identified by TPP at 52 °C. However, by lowering the temperature to 49 °C, the target identification performance of both non-ionic- and zwitterionic-detergent-based TPP remarkably improved and was comparable to the performance of detergent-free TPP at its optimal temperature (i.e., 52 °C). In addition, the kinase targets identified by zwitterionic detergent TPP at 49 °C are highly complementary to those identified by detergent-free TPP at 52 °C, indicating the different precipitation mechanisms in these two conditions. In fact, beyond heat, other external perturbances that can denature protein can also be applied for target protein identification; for example, our lab has successfully applied organic solvents [25], mechanical stress [26], and low pH [27] to induce protein precipitation for target protein identification. The combination of detergent and heat in this study can also be regarded as a hybrid external perturbance to denature proteins and thereby could be applied for target protein identification. However, the prerequisite for detergent TPP is that the detergent itself does not destroy the drug-binding pockets on proteins.

It is also worth mentioning that the centrifugation force applied during TPP analysis, after cell lysis or heating, may influence the extraction efficiency and recovery of membrane proteins. In this study, we utilized a centrifugation force of 20,000× *g*, both for proteome extraction and post-heating steps, as excessively high centrifugation forces can lead to the pelleting of membrane proteins, leaving them in the insoluble fractions.

In summary, our study provides guidance for temperature selection in TPP when detergents are applied. Additionally, we have demonstrated, for the first time, the potential of incorporating zwitterionic detergent in TPP for target protein identification. This novel combination of zwitterionic detergent and TPP enables the identification of target proteins in a manner that complements the detergent-free TPP method, expanding the possibilities for comprehensive target protein identification.

## 4. Materials and Methods

### 4.1. Chemicals and Materials

Staurosporine was purchased from Selleck. In this study, NP40 refers to IGEPAL, which shares the same formulation as NP40; however, NP40 is no longer available. IGEPAL, CHAPS, and all other reagents were purchased from Sigma (St. Louis, MO, USA).

### 4.2. Cell Culture

K562 cells were cultured in Iscove’s Modified Dulbecco’s Medium, abbreviated as IMDM (Gibco, Gaithersburg, MD, USA), containing 10% fetal bovine serum (FBS) (Gibco, NY) and 1% streptomycin (Beyond, Haimen, China) under the condition of 37 °C, 5% CO_2_. For harvesting the cells, the culture medium was discarded by centrifugation at 1000× *g* for 5 min. The collected cell pellets were washed three times with ice-cold PBS and stored at −80 °C for future usage.

### 4.3. Cell Lysis and Proteome Extraction

For PBS-TPP analysis, the frozen K562 cells were suspended in the original lysis buffer (PBS plus 1% EDTA-free protease inhibitor cocktail) followed by being snap-frozen in liquid nitrogen and being thawed in a 37 °C water bath three times. The resulting crude cell lysates were subjected to centrifugation at 20,000× *g*, 4 °C for 10 min to remove the cell debris. The supernatants were collected as cell lysate samples. The protein concentration was determined using a Bicinchoninic acid (BCA) assay (Beyotime, China) and adjusted to 4–5 mg/mL. For detergent-TPP analysis, the lysis buffer was composed of the original lysis buffer and 0.4% NP40, or 1% CHAPS.

### 4.4. Thermal Proteome Profiling (TPP) Analysis

The cell lysates were either incubated with 20 μM staurosporine (dissolved in DMSO, stock concentration of 2 mM) or DMSO at 25 °C for 30 min. After incubation, each mixture was distributed in parallel into four aliquots as four replicates for TPP analysis. The resulting samples were heated at 52 °C (or 49 °C or 47 °C) for 3 min in a thermal cycler followed by cooling at room temperature for 2 min. Subsequently, the samples were centrifuged at 20,000× *g*, 4 °C for 10 min to remove the precipitates. The collected supernatant proteins were denatured by adding a 3-fold volume of denaturing buffer (8M guanidine hydrochloride, 60 mM 4-(2-hydroxyethyl)-1-piperazineethanesulfonic acid (HEPES), pH 8.0). Subsequently, the protein samples were subjected to carbamidomethylation by adding 10 mM tris(2-carboxyethyl)phosphine (TCEP) and 40 mM chloracetamide (CAA) at final concentrations. The mixtures were then heated at 95 °C for 5 min. After cooling to room temperature, each sample was transferred to a 10 KDa filter unit (Vivacon^®^ 500 from Sartorius Stedim Biotech) to centrifugate at 14,000× *g* for 30 min to remove the denaturing buffer. The upper membrane of the filter unit was washed three times with 10 mM NH_4_HCO_3_. For detergent TPP, the upper membrane was first washed three times with 8M urea and 60 mM HEPES (pH 8.0) to remove the residual detergent, and then washed three times with 10 mM NH_4_HCO_3_. The proteins on the upper membrane were resuspended in 100 μL 10 mM NH_4_HCO_3_ and digested with trypsin (Promega) overnight. After digestion, the filter units were centrifuged at 14,000× *g* for 15 min to collect the peptides. The residual peptides were collected by washing the upper membrane with 100 μL 10 mM NH_4_HCO_3_ and centrifuging the filter unit at 14,000× *g* for another 15 min. The filtrates collected in the two cycles of centrifugations were combined and dried in SpeedVac (Thermo Fisher Scientific, San Jose, CA, USA).

### 4.5. SDS-PAGE Analysis

Cell lysates obtained under different lysis conditions (PBS, PBS supplemented with 0.4% NP40, or PBS supplemented with 1% CHAPS) were divided into 13 equal aliquots. Each aliquot was heated in a thermal cycler to a specific temperature (40 °C, 43 °C, 46 °C, 49 °C, 52 °C, 55 °C, 58 °C, 61 °C, 64 °C, 67 °C, 70 °C, 75 °C, and 80 °C) for 3 min followed by cooling at room temperature for 2 min. Subsequently, the sample was centrifuged at 4 °C for 10 min at 20,000× *g*. An equal volume of supernatants was collected and re-dissolved in reducing SDS-PAGE sample buffer and loaded onto the gel. SDS-PAGE quantifications were performed with ImageJ.

### 4.6. LC-MS/MS Analysis

The dried peptides were resuspended in 0.1% formic acid (FA). In the experiment for comparing extracted proteome, the peptides were analyzed by Q Exactive Orbitrap HF mass spectrometer coupled with a nano-flow HPLC system (Dionex Ultra 3000, Sunnyvale, CA, USA). The concentration of the peptide samples was determined by Nanodrop (Thermo Scientific, Waltham, MA, USA) and 1 μg peptide was automatically loaded onto a 3 cm C18 trap column at a flow rate of 5 μL/min. Peptides were separated on a 15 cm × 150 μm i.d. column (New Objective, PF360-75-10-N-5) packed in-house with 1.9 μm C18 ReproSil particles (Dr. Maisch GmbH). Binary buffers (A, 0.1% FA; B, 80% ACN, 0.1% FA) and liner gradients of 13–30% B for 80 min (on Q-Extractive-HF) were applied for peptide separation.

In the experiments for identifying staurosporine targets, the peptides were analyzed by Orbitrap Exploris 480 mass spectrometer coupled with a micro-flow HPLC system (Dionex Ultra 3000, Sunnyvale, CA, USA) and FAIMS interface. Specifically, 10 μg of peptides was separated on a commercial 15 cm × 1 mm i.d. column (ACQUITY UPLC Peptide CSH C18 Column, 130 Å, 1.7 μm; Waters). Binary buffers (A, 0.1% FA; B, 80% ACN and 0.1% FA) were used. Peptides were separated by linear gradients from 6% B to 32% B for 80 min followed by a linear increase to 45% B in 14 min at the flow of 50 μL/min. FAIMS parameters were set as follows: compensation voltage, −45 V; total carrier gas flow, 3.5 L/min.

For QE-HF, full MS scans were acquired between 350 and 1050 *m*/*z* at a resolution of 60,000 (AGC target of 3e6 or 20 ms maximal injection time). A total of 24 DIA segments on HF were acquired ranging from 410 to 970 *m*/*z* at a resolution of 30,000 (AGC target of 1e6 and auto maximum IT) with the normalized collision energy of 27. The first mass was fixed at 400 *m*/*z*.

For Exploris 480, full MS scans were acquired at 120,000 resolution (*m*/*z* = 200) spanning from *m*/*z* 350 to 1400 with the automatic gain control (AGC) target set to 3e6 and a maximum injection time (IT) of 45 ms. MS/MS scans were acquired in data-independent acquisition (DIA) mode with a resolution of 30,000 (*m*/*z* = 200). A total of 24 DIA segments were acquired, ranging from *m*/*z* 400 to 1000 at a resolution of 30,000 (AGC target of 2e6 and auto maximum IT), with the normalized collision energy (NCE) of 30%. The first mass was fixed at *m*/*z* of 300.

### 4.7. Data Processing

The raw mass spectra files were analyzed with Spectronaut (Biognosys AG, version 17) using the directDIA analysis module. A non-redundant Uniprot human database containing 20,185 proteins (downloaded in 2022) was imported as a FASTA file for directDIA searching. The digestion enzyme was set as “Trypsin”. The other parameters were set as default. For example, digestion type was set as “specific”; for modifications, “Carbamidomethyl(C)” was set as the fixed modification, and “Acetyl (Protein N-term)” and “Oxidation (M)” were set as variable modifications; for missing value imputation, “Global Imputing” was selected. The run-wise protein FDR and experiment-wise protein FDR were both set to the default values defined in Spectronaut, namely 0.01 and 0.05, respectively. Additionally, proteins present in the decoy were also removed. Protein quantification was performed at the MS2 level, and cross-run data normalization was achieved using a local normalization strategy [28] integrated within the Spectronaut software. The exported protein intensities were subjected to the Bayes *t*-test was performed between staurosporine- and DMSO-treated groups via R (limma package).

### 4.8. Bioinformatic Analysis

Principal component analysis (PCA) was performed using R (prcomp). Gene ontology (GO) cellular compartment analysis was performed on the DAVID platform (2022).

## 5. Conclusions

In this study, we investigated the effects of introduction of detergents on the performance of TPP in target protein identification. By investigating the target proteins of a pan-kinase inhibitor, we found that introduction of either non-ionic or zwitterionic detergent significantly decreased the number of identified kinase targets at 52 °C, the optimal temperature for soluble protein target identification. We further demonstrated that this may result from that detergents can decrease the thermal stabilities of proteins, making them more prone to precipitation. Guided by this speculation, we conducted experiments by lowering the applied temperature. Remarkably, we observed a significant improvement in the target identification performance of TPP in the presence of detergents, comparable to that without detergents. Our study underscores the importance of reducing the heating temperature in TPP when detergents are applied and also for the first time demonstrates that TPP is compatible with zwitterionic detergents for target protein identification, which could broaden the spectrum of target proteins identified through TPP. 

## Figures and Tables

**Figure 1 molecules-28-04859-f001:**
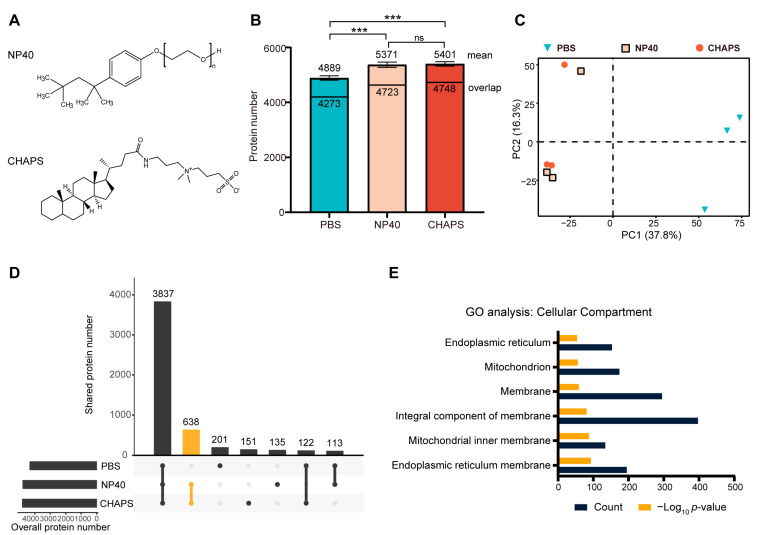
The introduction of detergents improved the coverage of the extracted proteome. (**A**) The chemical structures of the non-ionic detergent NP40 and the zwitterionic detergent CHAPS. (**B**) The numbers of identified proteins (three technical replicates) when K562 cells were extracted with PBS lysis buffer (composed of PBS and 1% proteasome inhibitor cocktail), PBS lysis buffer supplemented with 0.4% NP40, and PBS lysis buffer supplemented with 1% CHAPS. Mean and overlap refer to the average number of identified proteins and the number of proteins shared among three replicates. Significance levels obtained from *t*-tests using Prism (version 8.02) were encoded as *** *p* < 0.005; Ns denotes no significant difference. (**C**) PCA analysis showing proteomes that were extracted using detergents are clustered together. (**D**) The overlap of identified proteins in the corresponding dataset. (**E**) GO (gene ontology) cellular compartment analysis of 638 proteins that were present in both NP40 and CHAPS datasets but were absent in the detergent-free dataset. Membrane-related proteins are strongly enriched among these 638 proteins.

**Figure 2 molecules-28-04859-f002:**
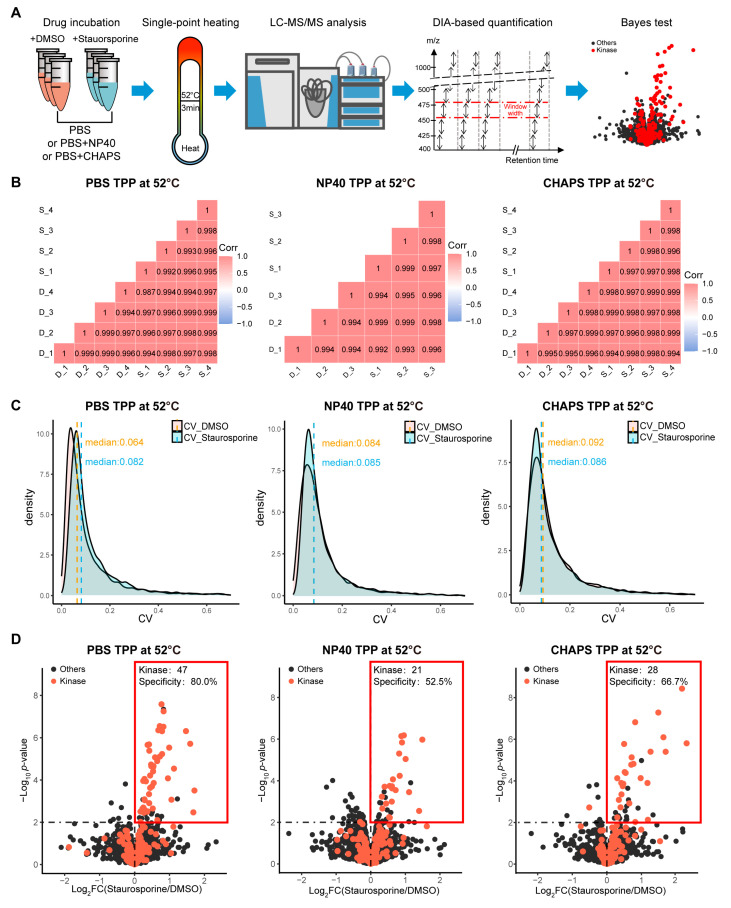
The introduction of detergents in TPP 52 °C substantially reduced the number of identified kinase targets. (**A**) The workflows of PBS-TPP 52 °C and detergent-TPP 52 °C experiments. (**B**) Peptide intensity correlation (Pearson) among replicates in the DMSO-treated group (D_1,2,3,4) and staurosporine-treated group (S_1,2,3,4). (**C**) Coefficient of variation (CV) for peptide intensity within each group, including DMSO-treated and staurosporine-treated groups. The median value of CV in each group is labeled. (**D**) Volcano plot visualization of staurosporine kinase targets identified by PBS-TPP at 52 °C (**left**), NP40-TPP at 52 °C (**middle**), and CHAPS-TPP at 52 °C (**right**). The kinase proteins are denoted with red points. The red frame denotes the significance cutoff, i.e., log_2_FC > 0 and −log_10_
*p*-value > 2. Specificity refers to the proportion of kinases present in the proteins that pass the significance cutoff.

**Figure 3 molecules-28-04859-f003:**
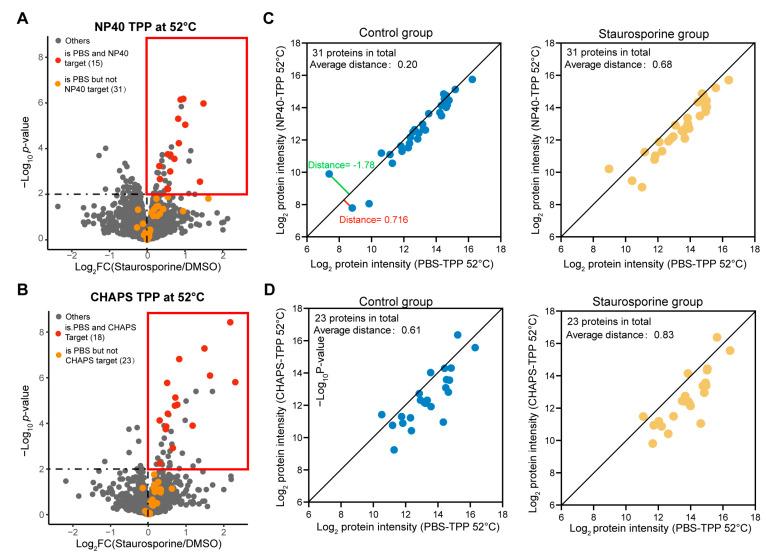
Characterization of the kinase proteins that were identified as staurosporine targets by PBS-TPP at 52 °C but not by NP40- or CHAPS-TPP at 52 °C. (**A**) Volcano plot visualization of NP40-TPP 52 °C dataset. 46 out of 47 kinase targets that were identified by PBS-TPP at 52 °C were also included in the NP40-TPP 52 °C dataset. (**B**) Volcano plot visualization of CHAPS-TPP 52 °C dataset. 41 out of 47 kinase targets that were identified by PBS-TPP at 52 °C were also included in the CHAPS-TPP 52 °C dataset. (**C**) **Left**: comparison of protein intensities in the control group between the PBS-TPP 52 °C dataset and NP40-TPP 52 °C dataset. **Right**: comparison of protein intensities in the staurosporine-treated group between the PBS-TPP 52 °C dataset and NP40-TPP 52 °C dataset. Distance > 0 denotes that the protein intensities in the NP40-TPP 52 °C dataset are lower than those in the PBS-TPP 52 °C dataset. The protein intensities are from the 31 kinase proteins that were identified as staurosporine targets by PBS-TPP at 52 °C but not by NP40-TPP at 52 °C. (**D**) As in (**C**) but the comparison is between PBS-TPP 52 °C dataset and CHAPS-TPP 52 °C dataset. The protein intensities are from the 23 kinase proteins that were identified as staurosporine targets by PBS-TPP at 52 °C but not by CHAPS-TPP at 52 °C.

**Figure 4 molecules-28-04859-f004:**
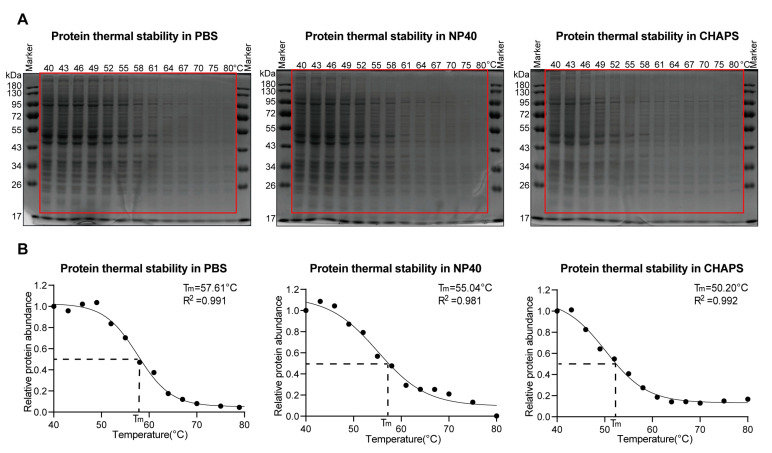
The presence of detergents decreased the thermal stabilities of the proteome. (**A**) SDS-PAGE readout of proteins that were collected from the supernatants after heating. From left to right: in the context of PBS (**left**), in the context of PBS supplemented with 0.4% NP40 (**middle**), and in the context of PBS supplemented with 1% CHAPS (**right**). (**B**) Normalized protein quantification value according to the area denoted by the red frame in (**A**). Tm represents melting temperature; R^2^ refers to the Pearson correlation between the actual values and the fitted values of the four-parameter logistic curve. The Tm curves were fitted using Prism (version 8.0.2).

**Figure 5 molecules-28-04859-f005:**
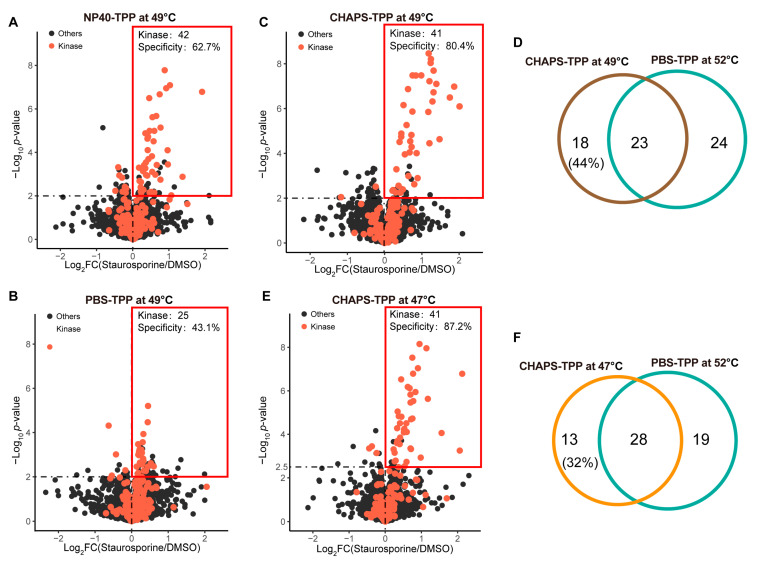
Lowering the applied temperature significantly improved the target identification performance of TPP when detergents were utilized. (**A**) Volcano plot visualization of staurosporine targets identified by NP40-TPP at 49 °C. (**B**) As in (**A**) but by PBS-TPP at 49 °C. (**C**) As in (**A**) but by CHAPS-TPP at 49 °C. (**D**) Overlap of kinase targets identified by CHAPS-TPP at 49 °C and PBS-TPP at 52 °C. (**E**) As in (**A**) but by CHAPS-TPP at 47 °C. (**F**) Overlap of kinase targets identified by CHAPS-TPP at 47 °C and PBS-TPP at 52 °C.

**Figure 6 molecules-28-04859-f006:**
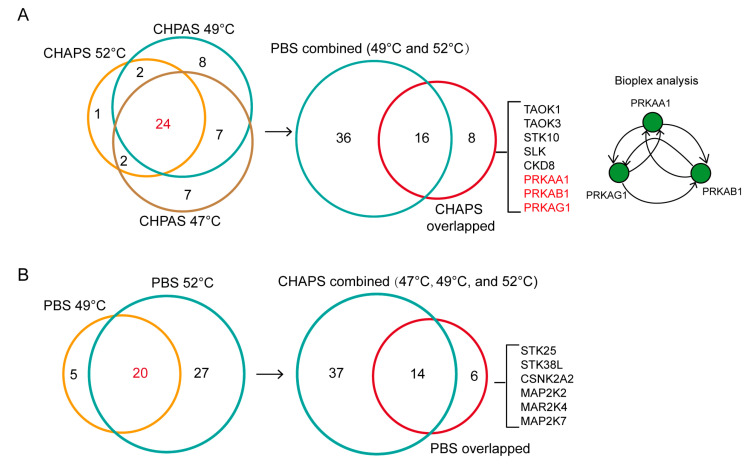
The unique kinase target proteins identified by CHAPS-TPP. (**A**) 24 target proteins that were shared in all CHAPS datasets are denoted as CHAPS overlapped. The number, 24, colored in red represents the number of CHAPS-overlapped kinase targets, corresponding to the red cycle on the right. Out of these 24 kinase target proteins, eight were exclusively identified as staurosporine targets in the CHAPS datasets in comparison to the PBS datasets. Among the eight proteins, Bioplex reported an interaction between three kinase target proteins: PRKAA1, PRKAB1, and PRKAG1. (**B**) 20 kinase target proteins that were shared in both two PBS datasets are denoted as PBS overlapped. The number, 20, colored in red represents the number of PBS-overlapped kinase targets, corresponding to the red cycle on the right. Out of these 20 kinase target proteins, six were exclusively identified as staurosporine targets in the PBS datasets in comparison to the CHAPS datasets. There are no Bioplex-annotated interactions among these six proteins.

## Data Availability

The mass spectrometry proteomics data have been deposited to the ProteomeXchange Consortium via the PRIDE partner repository with the dataset identifier PXD PXD042332.During the reviewing process, reviewers can access the data using the following ID: Username: reviewer_pxd042332@ebi.ac.uk. Password: RmZtVcG2.

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
