# Peer review of "The Introduction of Detergents in Thermal Proteome Profiling Requires Lowering the Applied Temperatures for Efficient Target Protein Identification"

_molecules, 2023, doi:10.3390/molecules28124859_

Round 1

Reviewer 1 Report

Thermal proteome profiling (TPP) is a method that is used for drug target protein identification. TPP is based on the concept that drug-bound target proteins are more resistant to heat-induced precipitation compared to unbound targets and consequently they will have a higher abundance in the soluble fraction of cells. The manuscript by Ye et al. investigates the impact of the use of a non-ionic detergent vs. a zwitterionic detergent on the optimal temperature for soluble target protein identification using TPP. The authors present data that can inform the selection of appropriate temperature ranges when detergents are used in studies incorporating TPP. The quality of the manuscript could be improved by addressing the following comments:

11. Use different wording to soften the tone of the statement in line 43: “TPP unarguably demonstrates excellent performance…”

22. Line 52: What is meant by “improve the proteome”?

33. Line 86: Add the standard deviations to the numbers of identified proteins from each condition.

44. Figure 1B lacks any assessment of statistical significance.

55. Line 88: “Principle component analysis” should be “Principal component analysis”

66. Figure 2: It is unclear why the authors chose an inconsistent number of technical replicates for the PBS TPP condition (n=4) vs. NP40 TPP (n=3) and CHAPS TPP (n=3).

77. Figure 2B: Please be consistent with the number of significant figures that are used for the correlation plots.

88. Presumably, the gel images in Figure S2 are included as technical replicates of the data shown in Figure 4A (line 200 – 202: “our SDS-PAGE results from two experimental replicates revealed a reduction in protein thermal stability of 2.6°C and 2.2 respectively…”) However, it is incorrect to consider these as technical replicates. The lower bound of the MW region used for densitometry analysis in Fig. S2 for the NP40 and CHAPS conditions was ~40 kDa, whereas it was ~20 kDa for the gels corresponding to these conditions in Fig. 4A.

99. Figure 4B: Consider changing the y-axis title from “Normalized gray value” to “Relative protein abundance”

110.  Volcano plots in Fig. 2D, 3A, 3B, 5A, 5B, 5C, 5E: Add vertical lines indicating the significant protein abundance fold-change cutoffs

111.  According to the information in Methods 4.3, the authors used a centrifugation speed of 20,000xg in their TPP analysis. In the Discussion section, the authors are encouraged to comment on the influence of centrifugation speed on the preferential pelleting of membrane proteins.

112.  Method 4.7: Details are lacking re: the data filtering criteria and the data normalization strategy.

Please correct the grammatical errors

Reviewer 2 Report

TPP is a relatively new technique for drug target identification. As noted in the manuscript, these targets are often membrane proteins that are insoluble and difficult to decipher with TPP. The use of detergents offers a possible solution to overcome this difficulty. The authors investigate the effect of a nonionic and a zwitterionic detergent on TPP.

I have some comments and suggestion to improve the manuscript:

1.       Please consider to change systemic to systematic in page 2, row 52.

2.       Figure 1: I recommend presenting the identified proteins on a Venn-diagram instead of a PCA plot and an overlapping diagram (Fig 1 C and D). I think PCA is not appropriate here.

3.       Figure S1 and its description on p 3 row 122-124. Does the whole proteome or the kinase proteome have the narrower distribution of thermal stability?

4.       p3 last sentence: please rephrase! e.g.: All experiments were performed in at least 3 (parallel) technical replicates.

5.       p5, row 150: The emphasis on excellent reproducibility is somewhat exaggerated. Technical repetitions and DIA analyzes were made. Reproducibility is due to the DIA method.

6.       p6 row 178. Please give a reference for the quantile method used for normalization.

7.       With the CHAPS-TPP a 3-member protein complex was identified (according to Bioplex). Shell we consider these proteins as real targets of the drug? Or just a false positive results of the effect of the detergent?  Is there any contradiction with p9 row 301?

8.       There are a lot of abbreviations in the manuscript which are not resolved. E.g.: CHAPS, IMDM, BCA, DMSO, TCEP, CAA…) Please define at first appearance all of them.

9.       p10 row 332: include “47°C “

10.   p10 row 335: Please change “carbamylated” to “carbamidomethylated”.

11.   p11: The instruments used are not specified completely. Please provide more details on the HPLC systems used (both nano-flow and micro-flow systems)

12.   Data processing: “The other parameters were set as default.” Please give more details or at least give a reference where these parameters can be found.

13.   „MS2 intensity of each protein was exported as protein intensity.” Please explain it and clarify what you mean by that.

14.    I propose to summarize the results in a (supplementary) table containing all identified kinases, indicating the experiments where they were found.
